# Hepatitis C Virus (HCV)-Ribonucleic Acid (RNA) As a Biomarker for Lymphoid Malignancy with HCV Infection

**DOI:** 10.3390/cancers15102852

**Published:** 2023-05-21

**Authors:** Yutaka Tsutsumi, Shinichi Ito, Souichi Shiratori, Takanori Teshima

**Affiliations:** 1Department of Hematology, Hakodate Municipal Hospital, Hakodate, 1-10-1, Minato-cho, Hakodate 041-8680, Japan; drterrakhan@blood-works.sakura.ne.jp; 2Department of Hematology, Hokkaido University Graduate School of Medicine, Sapporo 060-8638, Japan; s.shiratori@med.hokudai.ac.jp (S.S.); teshima@med.hokudai.ac.jp (T.T.)

**Keywords:** hepatitis C virus (HCV), Direct Acting Antivirals (DAA), Non-Hodgkin’s lymphoma (NHL), HCV-RNA, micro-RNA

## Abstract

**Simple Summary:**

Overview of the onset and progression of B-cell lymphoma under infection with the hepatitis C virus, and a review of the current status of biomarkers related to treatment efficacy and prognosis under the progress of DAA therapy.

**Abstract:**

The hepatitis C virus (HCV) is potentially associated with liver cancer, and advances in various drugs have led to progress in the treatment of hepatitis C and attempts to prevent its transition to liver cancer. Furthermore, reactivation of HCV has been observed in the treatment of lymphoma, during which the immortalization and proliferation of lymphocytes occur, which leads to the possibility of further stimulating cytokines and the like and possibly to the development of lymphoid malignancy. There are also cases in which the disappearance of lymphoid malignancy has been observed by treating HCV and suppressing HCV-Ribonucleic acid (RNA), as well as cases of recurrence with an increase in HCV-RNA. While HCV-associated lymphoma has a poor prognosis, improving the prognosis with Direct Acting Antivirals (DAA) has recently been reported. The reduction and eradication of HCV-RNA by means of DAA is thus important for the treatment of lymphoid malignancy associated with HCV infection, and HCV-RNA can presumably play a role as a biomarker. This review provides an overview of what is currently known about HCV-associated lymphoma, its epidemiology, the mechanisms underlying the progression to lymphoma, its treatment, the potential and limits of HCV-RNA as a therapeutic biomarker, and biomarkers that are expected now that DAA therapy has been developed.

## 1. Introduction

HCV is a factor in the pathogenesis of hepatocellular carcinoma. Compared with healthy individuals, HCV is associated with a 23 to 35 times greater rate of liver cancer occurrence [1,2]. Control of HCV is thus considered to be important to suppress the pathogenesis of hepatocellular carcinoma. Meanwhile, the mechanism of how HCV is involved in the development of hepatocellular carcinoma is still not clear [2]. There have been reports since 2000 suggesting the involvement of HCV in lymphoproliferative disorders [3,4]. The mechanism of pathogenesis to lymphoid malignancy in HCV-infected cases is gradually becoming clear. However, there are still many areas that remain unelucidated [5]. Meanwhile, concerning lymphoma complicated with HCV infection, reactivation of HCV when rituximab is used has been reported [6,7]. The recurrence of non-Hodgkin’s lymphoma (NHL) with increasing HCV-RNA load has also been reported [8]. These cases suggest the possibility that an increase or eradication of HCV-RNA may affect the progression of HCV-positive lymphoid malignancy and its course of treatment. In this review, we provide an overview of studies to date, as well as discuss HCV reactivation in HCV-infected patients, epidemiology of lymphoid malignancy, mechanism involved in its pathogenesis, the impact of DAA therapy on HCV-positive lymphoid malignancy, the potential and limitations of HCV-RNA as biomarker, and the potential of biomarkers that are needed in the DAA era.

## 2. HCV Reactivation When Rituximab Is Administered 

To trace the relationship between HCV and lymphoid malignancy, it is necessary to consider HCV reactivation during chemotherapy for HCV positive lymphoid malignancy. One model of the contribution to HCV reactivation is the decrease in B cells due to rituximab administration, which results in the decrease of antibody production and an increase in HCV load. Stamataki et al. reported cases of lysis of HCV-infected B cells when rituximab was used to treat cryoglobulinemia, resulting in the release of HCV and an increase in HCV viral load [9]. They posit that HCV loses its adherence to B cells when B cells are destroyed by rituximab administration, resulting in an increase in HCV viral load [10]. For chemotherapy of patients with B-cell NHL complicated with HCV infection, we have used rituximab alone or in combination with other drugs. In contrast to an increase in HCV viral load after rituximab administration, with chemotherapy alone, we have observed a decrease in HCV viral load after its increase, or a lack of increase in HCV viral load [11]. Our findings support the findings of Stamataki et al. However, it is difficult to explain the reason for the decrease in HCV viral load after its increase in peripheral blood when rituximab is not used. Because B cells do not recover for at least six to nine months after the use of rituximab, it is unlikely that B cells were reinfected [12,13]. It is very likely that the increased HCV in peripheral blood reenters the bloodstream and infects hepatocytes. As a result, cytotoxic T cells (CTL) are attacked, and hepatitis occurs [14]. On the other hand, there have been few reports of severe cases of hepatitis due to HCV reactivation. The reason is believed to be related to the fact that compared with the hepatitis B virus (HBV), HCV is more likely to become chronic. However, the mechanism remains unclear. It is known that after rituximab is administered, B cells not only decrease, but that changes in CD4- and CD8-positive T cells due to changes in cytokines and other factors also occur. As a result, CD8-positive T cells decrease, making it easier for HCV to proliferate like HBV. In the case of HBV, CD8-positive T cells are produced that target HBV antigens during CD8-positive T cell recovery. At the same time, memory T cells are impaired (they decrease). As a result, the phenomenon of HBV randomly attacking infected hepatocytes and causing hepatitis occurs. In such a case, the hepatitis tends to be more severe [15,16]. When HCV is reactivated, it is believed to produce hepatitis with a similar mechanism. However, it is known that HCV is not completely eliminated even though HCV-specific CTL in the host is produced [17,18]. These systems create escape mutation that allows HCV to slip past CTL recognition when HCV is reactivated after rituximab treatment, thus establishing immune tolerance to the host and promoting chronicity of HCV infection. At the same time, compared with HBV reactivation, severe hepatitis is prevented as a result.

Many studies on HCV reactivation predate the use of rituximab, and few have actually evaluated HCV reactivation on a large scale. There have been scattered reports of HCV reactivation and resulting hepatitis, but few reports of large-scale occurrences [7,11,19,20,21,22]. These reports include cases of death resulting from post-HCV activation hepatitis [19,21]. Although there have been few large patient studies, in a representative study, Ennishi et al. reported that of the patients who were HCV-positive, 27% had hepatitis, compared with 3% of patients who were HCV-negative; furthermore, HCV-positive patients treated with transaminase had a higher rate of hepatitis [7]. Arcaini et al. reported that liver damage was observed in 17.9% of HCV-positive patients when R-CHOP was used [23]. Together with Ennishi et al.’s report, these findings show that liver damage occurs in about 15–30% of HCV-positive patients. In these reports, fatal hepatotoxicity in HCV-positive cases and delayed treatment due to liver damage caused the exacerbation of lymphoma [24]. On the other hand, no need to delay treatment even with the occurrence of liver damage has been reported [23,25,26]. In addition, HCV reactivation when rituximab is used is considered to be more frequent with genotype 2 HCVI, although the number of cases is small [19]. There has also been a report that the fatality rate is higher in patients with high initial HCV-RNA load [6]. It may be that the severity of hepatitis and the ease of reactivation depend on the HCV genome and viral load. However, there is a possibility that debate about HCV reactivation may be resolved with the development of direct-acting antivirals (DAA).

## 3. Epidemiology of HCV-Infected B-Cell Lymphoma

Like HCV’s involvement in hepatocellular carcinoma, whether HCV is involved in lymphoproliferative disorders has also been studied. Cryoglobulinemia has been reported to be strongly associated with HCV [27,28,29,30,31]. Because cases of cryoglobulinemia are rare in Japan, and NHL has been found to be associated with HCV, this review will focus on B-cell NHL. There have been reports stating an association between HCV and B-cell NHL [31,32,33,34,35,36,37], and in existing reports, the rate of HCV-positive lymphomas is considered to be approximately 0.5-25% [8,38,39]. The rate is also considered to depend on the type of lymphoma. Marginal zone lymphoma (MZL), diffuse large B cell lymphoma (DLBCL), and lymphoplasmacytic lymphoma have been reported to have a high association with being HCV-positive [40]. Nieters et al.’s study found HCV-infected patients were more likely to have DLBCL and unclassifiable B-cell lymphoma [41]. Dai Maso et al. conducted a meta-analysis of 15 studies on the association between HCV infection and NHL and found a 2–2.5 relative risk of lymphomagenesis in HCV-positive cases. Similar trends were found for various subtypes of NHL [32]. Meanwhile, in Japan there have been few epidemiological studies of B-cell lymphoma in HCV-infected patients. Ohsawa et al. reported on six-year (average) follow-ups of 2,162 HCV-positive patients [42]. Four cases of NHL were found, and although the association did not reach the dangerous level of hepatocarcinogenesis, they found a moderate association. Recently, Alkrekshi et al. compared the rate of B-cell lymphoma in HCV-positive and HCV-negative patients based on a database of 72 million patients from 2013 to 2020. They reported that there were 940 cases of NHL in 129,970 patients in the HCV group versus 107,480 cases of NHL in 37,961,970 patients in the control cohort (odds ratio (OR) 2.6, 95%, confidence interval (CI) 2.4–2.7). A positive association was observed for chronic lymphocytic leukemia, follicular lymphoma, marginal zone lymphoma, lymphoplasmacytic lymphoma, diffuse large B-cell lymphoma, Burkitt’s lymphoma, non-Hodgkin’s T-cell lymphoma, and primary cutaneous T-cell lymphoma. There were no differences in mantle cell lymphoma. They also reported that the increased risk of HCV-associated lymphoma was persistent across genders, between Caucasians and African-Americans, and across age groups. While the risk of NHL in the HCV-negative population was higher in Caucasians than African-Americans (OR 1.8, 95% CI 1.7–1.8), the risk of HCV-associated NHL was not different [43]. The researchers also found that there were few cases of NHL in patients under 40 years of age. In a prospective cohort study, Rabkin et al. analyzed 95 HCV-positive cases where B-cell lymphoproliferative disorder has developed. They reported the development of lymphoproliferative neoplasia a mean of 21 years after HCV infection [44]. These above findings indicate that a certain period of time is required after HCV infection before lymphoproliferative neoplasia develops.

## 4. Mechanisms of B-Cell Lymphomagenesis in HCV-Infected Patients

Marcucci et al. reviewed the possibility of HCV-induced lymphoma in the research literature and hypothesized the following mechanisms: (1) growth of lymphoma due to antigenic stimulation, (2) suppression of tumor immunity due to HCV infection, (3) co-infection with an unknown tumor virus, and (4) direct tumor antigenicity of HCV [45]. We first became aware of the association between HCV and lymphoma in a case of HCV-positive DLBCL where rapid increase in HCV-RNA was observed prior to recurrence [8]. Subsequently, we conducted staining of lymphoma specimens from cases of HCV-positive lymphoma with HCV-specific antibodies, and found that nonstructural protein 3 (NS3), an HCV antigen, stained positive, both strongly and weakly, in 76.9% of the cases. However, there was no significant correlation between the degree of HCV staining and the rate of recurrence or resistance to treatment [46]. While these findings suggest the possibility that HCV is associated with lymphoma, because not all pathology specimens from HCV-positive lymphoma were positive for NS3, it is inferred that HCV may not be necessarily directly involved in lymphoma in HCV-positive cases.

A factor currently considered in the association of HCV with B-cell lymphomagenesis is the possibility of HCV infection of hepatocytes and lymphocytes due to the fact that both hepatocytes and lymphocytes share CD81 [47,48,49]. Furthermore, CD81 forms a stimulatory complex with CD19 and CD21, which results in the promotion of the activation and proliferation of B cells via the B-cell antigen receptor (BCR) [50]. In addition, CD81 upregulates chemokine receptor CXCR3, activating B cells [51].

On the other hand, intracellular viral replication is not necessarily required for tumorigenesis of those cells [52]. The reason is considered to be the possibility of the loss of viral genome from the nascent cell in the process of being inserted into the cell’s DNA or during cell replication [53]. Viral oncoproteins can also cause epigenetic dysregulation to genetically reprogram cellular gene expression. After determining these changes in the gene expression pattern, the viral genome may be lost completely. Thus, a hit-and-run mechanism may be sufficient to induce tumorigenesis of the host cell with the temporary acquisition of a complete or incomplete viral genome [53]. A hit-and-run mechanism has also been suggested for HCV, and some researchers have shown that in vitro, HCV can induce mutations in several genes associated with cellular replication, such as p53, bcl6, and beta-catenin [54,55]. However, the possibility that HCV produces phenotypes that increase mutation rates has not been confirmed in vitro or from lymphocytes obtained from chronically HCV-infected patients [56].

The involvement of chronic antigen stimulation in the pathogenesis of NHL has been shown in mucosa-associated lymphoid tissue (MALT) lymphoma, which arises from Helicobacter pylori (HP) infection [57]. It has been suggested that a certain period of time is required until lymphomagenesis [43,44]. It is believed that chronic antigen stimulation of B cells is mediated by CD81, causing oligoclonal expansion and finally monoclonal expansion of lymphocytes and leading to NHL pathogenesis. The HCV-E2 protein also causes the proliferation of B cells by activating the JNK pathway through binding to CD81 [51].

In addition, overexpression of anti-apoptotic protein bcl-2 is often observed in HCV-positive mixed cryoglobulinemia (MC). It is also considered to be a second hit for the transition of lymphocyte proliferation to lymphoma [58,59]. Interleukin 6 (IL6) has been reported to be involved in the transformation of MC to lymphoma. An increase in IL6 causes inflammation, bringing changes in host conditions, and may strongly stimulate tumorigenesis [60].

Thus, as surveyed above, mechanisms of the pathways of potential lymphomagenesis include active lymphocyte proliferation and replication by viruses, and associated with that, cytogenetic abnormalities; run-and-hit mechanism; and chronic antigen stimulation. These mechanisms may act singly or in combination to contribute to lymphoma pathogenesis.

## 5. Prognosis of HCV-Positive Lymphoma

There have only been retrospective analyses when it comes to examining the prognosis of HCV-positive B-cell malignant lymphoma and HCV-negative B-cell lymphoma. Some reports find poor prognosis of HCV-positive lymphoma [23,38], whereas other reports find no difference in prognosis [8,20] or good prognosis [25]. In these reports, the prognosis cannot be stated with certainty because of noticeable variations in the cases. For example, the good prognosis group had many young patients and patients with low-grade lymphoma and the poor prognosis group had many patients with high LDH [39]. In the retrospective study we conducted in 2011, we found that patients with HCV-positive lymphoma tended to have poor prognosis. However, the difference in prognosis was not significant because of the small number of cases [46].

Studying prognostic factors, Merli et al. analyzed prognostic factors of HCV-positive malignant lymphoma in 535 patients given an anthracycline-based therapy. They found that ECOG performance status of 2 or over, serum albumin below 3.5 g/dL, and HCV-RNA viral load over 1000 KIU/mL were significant prognostic factors. The researchers proposed a way to stratify patients into three risk categories with different overall and progression-free survival (low = 0; intermediate = 1; high-risk ≥ 2 factors) by combining the three prognostic factors into a new prognostic score [61]. 

Recently, Elbedewy et al. analyzed the prognosis of HCV-positive DLBCL in Egypt and reported that compared to uninfected cases, HCV infection was independently associated with poor prognosis [62]. They also reported that although it had been suggested that antiviral therapy may improve prognosis, it was not an independent prognostic factor. Regarding these prognostic factors, Zhang et al. reviewed and analyzed previous reports on HCV-positive NHL [63]. They found that the overall survival (OS) and progression free survival (PFS) were both significantly shorter for HCV-positive NHL, which also showed poorer response to treatment. HCV-positive NHL patients also exhibited an advanced disease stage, elevated LDH level, high-intermediate or high international prognosis index (IPI) and follicular lymphoma international prognosis index (FLIPI) scores, as well as spleen and liver involvement [63]. They also found that antiviral therapy against HCV improved OS and PFS, and furthermore, combination with rituximab led to good results for HCV-positive NHL. However, patients with low albumin levels and hepatic cirrhosis still had a poor prognosis [63].

Synthesizing the above findings, it can be considered that for HCV-positive NHL, prognosis is still poor compared to HCV-negative NHL unless antiviral therapy is provided. Reasons include the progression of hepatic cirrhosis, IPI and FLIPI risk factors, low albumin, and the maintenance of a certain level of HCV-RNA.

## 6. Antiviral Therapy with Interferon and HCV-Positive NHL

Concerning the prognosis of antiviral therapy for HCV-positive lymphoma, interferons initially dominated treatments. In 2002, Hermine et al. reported observing therapeutic efficacy in treating HCV-positive patients with splenic lymphoma with villous lymphocytes (SLVL) with interferon alone or in combination with ribavirin; in a case when lymphoma recurred, HCV-RNA was again detectable in blood [64]. Saadoun et al. further expanded this study to 18 patients with chronic HCV and SLVL as well as Type II mixed cryoglobulinemia (MC) (both symptomatic and asymptomatic). They were treated with interferon alone or in combination with ribavirin, and in 14 cases, a sustained complete hematologic response was achieved [65].

On the other hand, for low-grade lymphomas other than marginal zone lymphoma (MZL), while peg-IFN may have a better outcome than conventional IFN, most reports suggest a hematologic response of 60–77%, depending on the addition of ribavirin or not [66,67]. Arcaini et al. reported results of treating 134 HCV-positive low-grade lymphomas with interferon alone or in combination with ribavirin. The objective response rate (ORR) was 77%. Noteworthy were the findings that treatment of lymphoma had greater efficacy in patients who achieved sustained virological response (SVR) and that there was no difference in therapeutic outcome between MZL and non-MZL non-grade lymphoma. Furthermore, 34 patients who received second-line antiviral therapy also had similar results [68].

Meanwhile, concerning DLBCL, we examined pre- and post-treatment changes in HCV-RNA and found that patients with lower HCV-RNA levels after treatment levels were less likely to relapse, whereas those with higher HCV-RNA levels after treatment were more likely to relapse or be refractory to treatment. In particular, patients that become HCV-negative after interferon therapy for malignant lymphoma did not relapse [46]. Suppression of HCV with interferon may thus contribute to a better prognosis of HCV-positive DLBCL. However, it has been reported that antiviral therapy with INF or ribavirin for HCV did not show positive results for immunochemotherapy, such as standard Rituxan or chemotherapy. This is believed to be due to the aggressiveness of DLBCL, which exceeded therapeutic efficacy as a result of gaining additional mutations and other factors [69].

The prognostic value of treatment with antiviral drugs for HCV after treatment for DLBCL was studied with retrospective analysis and combined analysis of retrospective prospective occurrences. In the group with antiviral therapy, 5-year OS and PFS rates may both improve [70,71]. However, Michot et al. reported the possibility that some cases of DLBCL may have been transformed from splenic marginal zone lymphoma (SMZL). In Europe and the U.S., there are many cases where SMZL is associated with HCV. These cases are highly responsive to HCV antiviral therapy, suggesting the possibility of an improved prognosis. On the other hand, Michot et al.’s report included some patients in the antiviral therapy group who did not achieve SVR, and Hosry et al.’s report included many patients with hepatic cirrhosis in their analysis. These factors may have adversely affected OS and PFS rates [70,71]. A factor that makes the treatment of hepatic cirrhosis difficult is Rho-associated kinase 2 (ROCK2), which is involved in hepatic fibrogenesis [72]. It has been suggested that besides promoting liver fibrosis, ROCK2 is also involved in the progression of lymphoma [73]. This suggests that hepatic fibrogenesis creates an environment conductive to lymphoma growth, which makes treatment difficult. On the other hand, in a retrospective study, La Mura et al. reported the group that received antiviral therapy did not experience recurrence and that antiviral therapy was associated with longer DFS [74].

The above findings suggest that in HCV-positive DLBCL patients, antiviral therapy with INF by itself does not achieve sufficient results; instead, side effects are noticeable. On the other hand, adding this therapy after immunochemotherapy may contribute to better disease-free survival (DFS) and OS rates.

## 7. Relationship between Hepatitis C and Lipids

Chronic HCV infection has been known to cause not only liver fibrosis and hepatomas but also fatty liver and disorders of lipid metabolism. Advances in experimental systems that can evaluate the HCV infection cycle have revealed that HCV exploits the lipid metabolic system with a variety of steps to enable efficient replication in hepatocytes [75,76,77,78,79]. In addition, it has been discovered that scavenger receptor type 1 (SR-B1) and low-density lipoprotein receptor (LDLR), which are receptors necessary for HCV entry, are also, respectively, HDL and LDL receptors [80,81,82,83]. Furthermore, it has been discovered that the lipid droplet is used as scaffold in the production of HCV core particle [84], and that apolipoproteins play an important role in the production of HCV infectious particles in the ER lumen [85,86,87]. Thomssen et al. first found that lipoproteins bind directly to HCV particles in sera of patients with chronic hepatitis C [88]. HCV particles interacting with lipoproteins are called lipoviroparticles (LVPs). LVPs with specific gravity close to that of LDL and VLDL are highly infectious, suggesting that HCV particles’ interaction with lipoproteins contribute to HCV’s high infectivity [89,90,91]. Furthermore, when the lipid composition of purified viral particles was analyzed, it was found that it was similar to the composition of VLDL and LDL [92]. Thus, HCV and lipids are deeply involved in infectious proliferation. Statins, which inhibit HMG-CoA reductase, have been shown to inhibit HCV genome replication in in vitro evaluation. In clinical trials, statins have been found to have anti-HCV activity in combination with other drugs [93,94,95]. Considering that HCV hijacks the lipid metabolic system, drugs that inhibit lipid metabolism may also inhibit HCV replication, and hold promise for DAA-resistant cases.

## 8. Treatment of HCV-Positive NHL in Era of Interferon-Free DAA Changes in DAA Therapy

HCV is currently classified into six genotypes, with genotypes 1 and 2 each having two subtypes (1a, 1b and 2a, 2b). In Japan, genotypes 1b, 2a, and 2b are predominant. With the exception of some DAAs agents, the antiviral efficacy of IFN and DAA differs depends on the genotype, so determining the genotype is essential for treatment. In 2013, the HCV NS5B nucleotide polymerase inhibitor sofosbuvir (Sovaldi) and ribavirin combination (administration of 12/24 weeks) was approved for genotype 2 and 3 chronic hepatitis and compensated cirrhosis, the first treatment for such HCV conditions in the world. In 2014, the NS5B-NS5A inhibitor combination of ledipasvir/sofosbuvir (Harvoni) (12-weeks) was approved as first-line treatment for genotype 1 chronic hepatitis and compensated cirrhosis, with a reported SVR12 of 95%. In Japan, the same treatment method was approved in 2015, and it showed outstanding therapeutic efficacy [96]. However, while the SVR12 in patients without Y93 or L31 resistance mutations was as high at 98%, in patients with Y93 mutations or Y93 + L31 double resistance mutations, the SVR of the ledipasvir/sofosbuvir combination was found to be around 90%. In 2017, a new combination of NS3-NS5A inhibitor glecaprevir/pibrentasvir (8 and 12 weeks) (Mavillet) was approved. The drug showed high therapeutic efficacy in patients with Y93+L31 double resistance. The combination therapy achieved more than 98% SVR12 after eight weeks of treatment for first-time treatment of genotype 1 and 2 chronic hepatitis [97] and shortened the treatment period from the heretofore 12 weeks of DAA treatment. Furthermore, a high therapeutic efficacy of more than 90% was achieved with 12 weeks of treatment in patients with prior DAA treatment failures. However, the glecaprevir/pibrentasvir combination showed poor therapeutic efficacy for patients with P32 deletion, which occurs in about 5% of patients with prior DAA treatment failures (mainly patients for whom DCV/ASV combination treatment was not effective). Next, the combination therapy velpatasvir/sofosbuvir (12/24 weeks) (Epclusa) was approved. This treatment provides a new option for patients with decompensated cirrhosis (12-week treatment) and patients with prior failed DAA treatments and P32 deficiency (24-week treatment with the addition of ribavirin) [98]. Other drugs have also been developed around the world and are being used differentially depending on patient conditions such as the presence of hepatic cirrhosis, renal failure, and maintenance dialysis [99].

## 9. Significance of DAA for HCV-Positive Indolent Lymphoma

For indolent NHL, it has been reported that only antiviral therapy improved splenic marginal zone lymphoma and marginal zone lymphoma, where subcutaneous “lipoma-like” nodules are formed. Antiviral therapy by itself is expected to improve prognosis of HCV-positive lymphoma [100,101,102].

There are still only a few comprehensive reports on the efficacy of DAA therapy for HCV-positive indolent NHL. Representative studies are shown in Table 1. Arcaini et al. used DAA therapy for HCV-positive indolent NHL patients and reported that SVR was achieved in 45 of 46 patients (including 7 with hepatic cirrhosis); one patient failed to continue treatment. They reported that the treatment for lymphoma resulted in achieving complete remission (CR) or partial remission (PR) in 27 or 37 patients with MZL and in 31 of 46 patients with other forms of indolent NHL. The cases of progressive disease (PD) in the report were hepatic cirrhosis that transformed to DLBCL. Two other cases of MZL became PD [103]. Mereli et al. reported similar results, with an SVR of 92.5% with DAA. The results of treatment for lymphoma were CR+PR in 13 of 27 patients with MZL and in 18 of 40 patients with other indolent NHL. The researchers reported that two patients died of lymphoma exacerbations, one patient died of a cause unrelated to lymphoma, and three patients had exacerbations in re-staging. In the report, being a male patient and the white blood cell count were associated with lymphoma exacerbations [104]. Both reports above give promising results in terms of OS and PFS rates and suggest that the introduction of immunochemotherapy in the early stage may not be necessary for patients with HCV-positive indolent NHL who have some hematologic response after DAA therapy. On the other hand, there were also results showing that compared with cases of HCV-positive patients treated also with interferon mentioned above, the effects may be slightly inferior. These results indicate the undeniable anti-tumor effects of interferon itself. Furthermore, since some cases of disease progression were observed at a relatively early stage, it is hoped that the poor prognosis factors shown by Mereli et al. will be confirmed in the majority of studies. Frigeni et al. compared cases of HCV-positive indolent lymphoma treated with DAA, including Arcaini et al.’s cases and previous cases, with cases of indolent lymphoma treated with DAA that included IFN. They found that DAA therapy had superior outcome for SVR and duration of treatment, and treatment that included IFN had superior CR rate. These findings indicate the possibility, again, that IFN may have antitumor effects [105]. The researchers also found no CR or PR in CLL patients, suggesting that there are some types of lymphomas that require treatment of the lymphoma itself, regardless of SVR. Based on the above findings, for patients with HCV-positive indolent NHL who are considered to have poor prognosis, and for those who have SVR against HCV but who do not respond adequately to treatment alone, the therapeutic efficacy of a combination of drugs that includes anti-CD20 antibody, BTK inhibitor, BCL2 inhibitor, and lenalidomide is expected to be promising. The reactivation of HCV amid the state of immunosuppression by these drugs is also of interest.

## 10. Significance of DAA in Aggressive HCV-Positive Lymphoma

Table 2 shows the results of first-line DAA therapy against mainly HCV-positive DLBCL. We evaluated the prognostic value of antiviral therapy against HCV after remission was achieved with CHOP or CHOP-like therapy combined with rituximab in five successive cases of HCV-RNA-positive DLBCL. The control groups consisted of a group of HCV-RNA-positive DLBCL cases prior to this trial (control 1), and a group of cases that tested negative for HIV, HCV, and HBV (control 2). All the cases were in remission at the time of initial treatment. The results showed that the DAA group had more genotype 2 patients, and the control 1 group had more genotype 1 patients. Although the genotype difference may have influenced the results, all five patients who received DAA therapy after immunochemotherapy survived without any recurrence at the two-year mark after treatment [106]. However, after the completion of this study, one patient who had also been infected with HBV from the time of initial onset developed hepatic cirrhosis, recurred DLBCL, and died. While there is a question of whether DAA therapy should be given concurrently or sequentially, in a report where immunochemotherapy (I-CT) and DAA therapy were performed, neutropenia and grade 3 fever associated with neutropenia were observed, but almost all patients were treated as planned [107,108]. A report found that low IPI risk scores and antiviral therapy were factors correlated with better prognosis [107]. Meanwhile, another study found that an IPI score of high-intermediate risk or above, or the presence of two or more extra-lymphatic node invasions, was considered to be the predominant factor for poor prognosis [109]. Further analysis of these factors with a large number of cases is desired.

Merelli et al. studied salvage therapy on patients with relapsed or refractory HCV-positive DLBCL with concurrent or sequential DAA therapy. They reported that while four patients died, the 4-year overall survival (OS) rate was 76% [110]. This result suggests that DAA therapy for HCV-positive DLBCL can be carried out concurrently. Even when carried out sequentially, it is considered to have a positive effect on HCV-positive DLBCL, such as during recurrence. The findings indicate that DAA therapy to suppress HCV at any point in time is important (although it should be conducted while the therapeutic efficacy of I-CT can be expected).

Recently, a retrospective study reported on cases where DAA therapy was performed or not performed for HCV-positive mantle cell lymphoma. Of the ten patients who did not receive DAA, eight had PD or relapse, and seven of them died. On the other hand, three of the four patients who received DAA maintained CR, and all four were reported to have survived at the time of the report. These results reaffirm the importance of DAA for HCV-positive NHL [111].

## 11. HCV-RNA as Biomarker

The above results suggest that as a result of HCV-RNA loss due to DAA and other means, therapeutic efficacy for HCV-RNA-positive NHL and its prognosis are improved. This suggests that HCV-RNA is useful as a biomarker for NHL. It also brings up the question of whether HCV-RNA is useful as a biomarker for NHL recurrence. We have previously reported on a case of NHL recurrence with increased HCV-RNA load [8]. However, this case occurred prior to the introduction of DAA, and is uncharacteristic in light of current conditions. Although there have been few cases of HCV-positive NHL after the introduction of DAA, what is noteworthy are scattered reports of recurrence immediately after SVR is achieved with DAA. In these reports, recurrences of HCV-positive DLBCL and other lymphomas occurred after I-CT and DAA; furthermore, a rise in HCV-RNA level was not detected [112,113,114,115]. These reports concerned NHL recurrences relatively early after SVR was achieved by DAA, so the involvement of DAA is suspected. On the other hand, we have encountered a case of NHL recurrence four years after SVR was achieved in an HCV-positive NHL patient who had undergone DAA therapy and achieved remission, even though SVR was maintained [106]. It is thus not clear whether DAA is directly involved in recurrences of HCV-positive NHL. In the current situation, where antiviral therapy is being applied with DAA, because recurrence of NHL has been observed despite SVR maintenance, an increase or decrease in viral load may not be appropriate as a biomarker for recurrence. In the past, attempts have been made to identify serum biomarkers for the presence or absence of B-cell NHL in HCV-RNA-positive cases, and sCD27, sIL-2Rα, gammaglobulins and C4 levels associated with the presence of overt B-NHL in HCV-infected patients have been reported [116]. The study was also conducted before the introduction of DAA, so it is unclear whether HCV-RNA can be used as a biomarker for HCV-positive NHL when SVR has been achieved with DAA. Future investigation of this area would be beneficial.

## 12. Micro RNAs (miRNAs) as Biomarkers

MicroRNAs (miRNAs) are single-stranded RNA molecules, approximately 21–25 nucleotides long. They are involved in the post-transcriptional regulation of gene expression in eukaryotes. The human genome is believed to encode more than 1000 miRNAs. miRNAs bind to their target mRNAs with incomplete homology. In general, they bind to the 3′ UTR of the mRNA of the target gene to destabilize it and suppress protein production through translational downregulation. miRNA-mediated transcriptional repression plays an important role in a wide range of biological processes, including development, cell proliferation and differentiation, apoptosis, and metabolism [117]. These various miRNAs are known to activate both canonical and alternative NF-κB pathways [118]. Of these miRNAs, miR-26b is known to be involved in lymphomagenesis through downregulation, and this phenomenon has been observed in HCV-positive splenic marginal zone lymphoma (SMZL) [119]. Furthermore, it has been reported that with regard to NHL, in addition to a decrease in miR-26b, an increase in expression of miR-21, mi-R16, and miR-155 was detected [120]. However, these miRNAs are associated with the growth and pathogenesis of HCV-positive lymphoproliferative disorders. Their abnormal expression and repression may become unchecked when SVR is achieved by DAA. Their usefulness as biomarkers in the DAA era is thus unclear. On the other hand, with the advancement of DAA, HCV-RNA is not necessarily useful as a biomarker. Considering that there have been cases of recurrence immediately after SVR by DAA, biomarkers that can, from the outset, stratify cases with poor prognosis are deemed necessary. Augello et al. analyzed lymph nodes obtained from 19 patients with HCV-positive DLBCL, 18 patients with HCV-negative DLBCL, 30 patients with HCV-positive reactive lymph nodes, and 30 patients with HCV-negative reactive lymph nodes. They found decreased expression of miR-138-5p and increased expression of miR-147a, miR-147b, and miR-511-5p in HCV DLBCL to be factors of poor prognosis for HCV-positive DLBCL patients. These miRNAs hold promise as biomarkers that can stratify poor prognostic cases from the outset for HCV-positive NHL patients [121]. It is desirable to study if these biomarkers are still useful in the current era of DAA advancements.

## 13. Conclusions

As a result of advances in DAA therapy, the prognosis of HCV-positive lymphoproliferative disorders has improved. The reduction or loss of HCV-RNA is useful as a biomarker predicting the success of the treatment of an HCV-positive lymphoproliferative disorder or its prognosis. However, HCV-RNA loses its usefulness for predicting recurrence. On the other hand, in some lymphoproliferative disorders where SVR was achieved by DAA, early recurrence was observed. There is thus a need to stratify lymphoproliferative disorders with possible poor prognosis from the outset. Some miRNAs are biomarkers that can predict lymphoproliferative diseases with a poor prognosis from an early stage. Their usefulness after SVR is achieved by DAA is being studied and promises to bring about further practical applications.

## Figures and Tables

**Table 1 cancers-15-02852-t001:** Previous studies of low-grade lymphoma treated with DAAs.

	Disease	n	Treatment Outcome				
			SVR	CR	PR	SD	PD
Arcaini et al. [80]	MZL	37		11	16	6	4
	CLL/SLL	4		0	0	4	0
	FL	2		0	2	0	0
	LPL	2		0	1	1	0
	Low grade NOS	1		1	0	0	0
	Total	46	45	12	19	11	4
Mereli et al. [81]	MZL	27		7	6	10	4
	CLL/SLL	2		0	1	0	1
	FL	1		0	1	0	0
	LPL	6		0	1	5	0
	Low grade NOS	4		1	1	1	1
	Total	40	38	8	10	16	6

Abbreviations: DAA: direct-acting antiviral agents; SVR: sustained virological response; CR: complete remission; PR: partial remission; SD: stable disease; PD: progressive disease; MZL: marginal zone lymphoma; CLL/SLL: chronic lymphocytic leukemia/small lymphocytic lymphoma; FL: follicular lymphoma; LPL: lymphoplasmacytic lymphoma.

**Table 2 cancers-15-02852-t002:** Previous studies of aggressive lymphoma treated with DAAs.

	Disease	n	DAA		Treatment Outcome				
			Concurrent	Sequential	SVR	CR	PR	PD	Died
Tsutsumi Y et al. [83]	DLBCL	5	0	5	5	4	0	1	1
Persico M et al. [84]	DLBCL	20	20	0	19	19	0	unknown	1
Occhipinti V et al. [85]	DLBCL	7	7	0	7	7	0	0	0
Mereli M et al. [86]	DLBCL	45	9	36		42	0	3	2
	G3 FL	2	0	2		2	0	0	0
	Total	47	9	38	45	38	0	3	2

Abbreviations: DAA: direct-acting antiviral agents; SVR: sustained virological response; CR: complete remission; PR: partial remission; PD: progressive disease; DLBCL: diffuse large B-cell lymphoma; FL: follicular lymphoma.

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
