# Peer review of "Hepatitis C Virus (HCV)-Ribonucleic Acid (RNA) As a Biomarker for Lymphoid Malignancy with HCV Infection"

_cancers, 2023, doi:10.3390/cancers15102852_

Round 1

Reviewer 1 Report

This is a Review and brings the Reader up to speed with the newest literature in the field. I enjoy reading the Manuscript and it gives a thorough treatment and review of the literature. It also describe some cases which have been the patients of the Authors themselves. HCV is still a severe disease and difficult to treat. Even though we have ever better medicines, the changes to best practice and mechanisms of new drugs, require constant review. This is a timely review written by current experts in the field. 

I would recommend to include a new section on HCV and lipids. The association of HCV with lipoproteins is firmly established and lipids are important both during HCV RNA synthesis and virus assembly as well as during virus uptake. If the Authors could write a new section on the importance of patient lipid in cancer development and treatment, that would increase the scope of the Manuscript.

This is a Review and brings the Reader up to speed with the newest literature in the field. I think the English is very good. It is well written and easy to understand.

Author Response

  1. Added section on lipids and HCV.

Reviewer 2 Report

This is a review article on Hepatitis C virus (HCV)-Ribonucleic acid (RNA) as a biomarker for lymphoid malignancy with HCV infection. The authors finally concluded to need to stratify lymphoproliferative disorders with possible poor prognosis from the outset. Some miRNAs are biomarkers that can predict lymphoproliferative diseases with a poor prognosis from an early stage. It is important to consider the relationship between HCV and lymphoid malignancy. So this review is interesting and important. 

1) The authors state that The reduction and eradication of HCV-RNA by means of DAA is thus important for the treatment of lymphoid malignancy associated with HCV infection, and HCV-RNA can presumably play a role as a biomarker. The eradication of HCV-RNA is important for the treatment of hepatitis; a decrease does not mean much. Rather than a reduction, a better interpretation is that the original HCV-RNA is low. 

2) The authors describe that cases of progression of HCV-positive cases to hepatic cirrhosis without the occurrence of hepatocellular carcinoma have been observed. HCV infection eventually results in severe liver disease manifesting as advanced fibrosis, cirrhosis, and hepatocellular carcinoma. Since HCV progresses in stages, it is not surprising that there is cirrhosis without HCC. 

3) Fibrosis is one of the most important factors in the development of HCC, but is fibrosis involved in the development of malignant lymphoma? You have described that it is related to prognosis, but is it related to the development of the disease?

Author Response

  1. As the Reviewer pointed out, we consider the eradication of HCV-RNA to be important and have revised the text accordingly.
  2. We have revised the text in response to the point raised by the Reviewer.
  3. ROCK2 has been observed to be involved in hepatic fibrogenesis. Furthermore, studies have raised the possibility of ROCK2’s involvement in the progression of lymphoma and the possibility of treating lymphoma by targeting ROCK2. We have added text on possible links between hepatic fibrogenesis and ROCK2 and lymphoma.

Round 2

Reviewer 2 Report

It was a polite correction. 

I found your reference to Rho-associated kinase 2 (ROCK2) helpful.

Thank you for your information.